# How Well Does the CWEQ II Measure Structural Empowerment? Findings from Applying Item Response Theory

**Farinaz Havaei * and V. Susan Dahinten**

School of Nursing, University of British Columbia, Vancouver, BC V6T 2B5, Canada;
Susan.Dahinten@nursing.ubc.ca
* Correspondence: farinaz.havaei@ubc.ca or farinazhavaei@gmail.com

**Abstract:** The main purpose of this paper is to examine the psychometric properties of the original five-point CWEQ II using Item Response Theory (IRT) methods, followed by an examination of the revised three-point CWEQ II. (1) Background: The psychometric properties of the CWEQ II have not been previously assessed using more robust techniques such as IRT. (2) Methods: This is a secondary analysis of baseline data from 1067 staff nurses whose leaders had attended a leadership development program. Data were analyzed using a polytomous IRT model. (3) Results: The two versions of CWEQ II fit the SE data equally as each had only one poor-fitting item. For the five-point CWEQ II, discriminant ability was poor for a majority of the items; one item demonstrated a disordinal step difficulty parameter; and item reliability was supported for a relatively wider range of SE levels. The discriminant ability and reliability of items for the three-point CWEQ II was better than those of the five-point CWEQ II, but for a narrower range of SE levels; and the disordinal step difficulty parameter was resolved. (4) Conclusion: The appropriate use of each version of the scale depends on the conditions of the work setting targeted.

**Keywords:** Item Response Theory; structural empowerment; CWEQ II; psychometric testing

## 1. Introduction

Structural empowerment (SE) is a widely-used concept in the nursing literature, one that is often considered when funders and nurse administrators are making important resource decisions such as offering nursing employees educational or promotional opportunities. Structural empowerment is defined as one's perceptions of access to information, resources, opportunities, support, and formal and informal chains of power [1]. In the nursing context, outcomes predicted by SE include psychological empowerment [1], burnout [2], job satisfaction [3], organizational commitment [4], turnover [5] and quality of nursing care [6].

When employees perceive they have access to empowering structures, they are more likely to identify their workplace as a healthy work environment. Healthy workplace environments are known as key determinants of positive nurse and patient outcomes. Aiken and colleagues studied the effects of unhealthy work environments caused by the extensive health care restructuring and downsizing that had occurred internationally over a decade ago. These environments were shown to compromise nursing care, as evidenced by increased nurse burnout, intent to leave, adverse events, and patient morbidity and mortality [7–10]. Such findings prompted Dr. Heather Laschinger and nursing colleagues from University of Western Ontario to attend to the notion of SE, leading to the development of the Conditions of Work Effectiveness Questionnaire I (with 58 items) and the shorter

II (CWEQ I and CWEQ II). The CWEQ II has been studied and used frequently in nursing research since 2000 [11].

The CEWQ is a 19-item measure with 6 subscales: opportunities (e.g., the extent to which one has the opportunity to gain new skills in his/her job), information (e.g., the extent to which one has access to information related to the current state of the hospital), support (e.g., the extent to which one is provided with problem solving advice), resources (e.g., the extent to which one has time to do job requirements), informal power (e.g., the extent to which one is sought by work peers), and formal power (e.g., the extent to which one is rewarded for innovation in his/her role) [11]. Items are rated on a five-point Likert-type response scale with scores ranging from 0 (*strongly disagree*) to 4 (*strongly agree*). With the exception of the informal power subscale, each subscale consists of three items. Mean subscale scores are summed to obtain a total SE score, with higher total scores indicating higher levels of SE.

There has been extensive evaluation of the CWEQ II. For example, a systematic review that was limited to psychometric studies of SE and psychological empowerment identified seven studies that evaluated the scale's content, convergent, discriminant, predictive, concurrent, and factorial structure validity and reliability [12]. Other more recent assessments of the instrument have been focused on its cultural adaptation [13–17]. Although there has been extensive examination of the psychometric properties of the CWEQ II, these psychometric studies have relied exclusively on Classical Test Theory (CTT); the CWEQ II has not been previously assessed using more robust techniques such as Item Response Theory (IRT). Item Response Theory is a probabilistic model that uses item characteristics and one's pattern of responses to estimate one's overall level of the latent variable in question (e.g., perceptions of SE) [18]. This is contrary to CTT methods where item characteristics are not taken into consideration. Item Response Theory also allows us investigate the psychometric implications of using different response scales. The main purpose of this paper is to examine the psychometric properties of the original five-point CWEQ II (with a response scale of 5 categories) using IRT methods followed by an examination of the revised three-point CWEQ II (with a response scale of 3 categories) using a large sample of staff nurses from British Columbia, Canada.

Item Response Theory addresses three shortcomings associated with the CTT procedures. First, rather than assuming that all items in the scale have the same characteristics (e.g., the same level of difficulty), IRT methods account for potential differences between item characteristics when estimating respondents' ability levels (i.e., level of the construct in question). Second, whereas CTT is predominantly concerned with the psychometric characteristics of overall scale or subscales, IRT focuses on item level analysis. Third, CTT methods provide a single estimate of the scale's measurement precision (i.e., reliability) which applies to all items and all respondents. The disadvantage of single index of reliability is that reliability tends to be lower when an individual's ability level is at the extreme ends of the spectrum [18,19]. In contrast, IRT methods assess the measurement precision of each item, for individuals across a continuum of abilities.

The application of IRT analysis to CWEQ II provides us with additional information about the scale, unobtainable through CTT. This information can facilitate more accurate measurement of SE, better guide nurse leaders' decision making with respect to the workplace environment, and provide direction towards leadership strategies that enhance SE. The specific research questions addressed by this study were as follows: (a) Does an IRT model fit SE data obtained from staff nurses? (b) Does evidence support reliable and valid interpretation of SE scores? and (c) How does validity and reliability change with different number of response options? First, an overview of IRT, in particular polytomous IRT as it was applied to examining the study data is provided, followed by a description of the underpinning assumptions and most common issues in IRT.

## 2. An Overview of IRT

Item Response Theory uses complex mathematical modeling based on the idea that the probability of selecting a 'correct' response for an item is dependent on one's 'ability level' and item characteristics

such as 'item difficulty' [20]. For the CWEQ II, with a Likert-type response scale where there is no correct response, this means that the probability of selecting a particular response (e.g., *strongly agree*) would be dependent on one's overall level of SE (i.e., ability) and item difficulty. Item difficulty pertains to the level of ability that would be required to endorse the correct or, in this case, the higher response option. For example, for 'easy' items, even individuals with lower levels of SE would *strongly agree*. On the other hand, for 'difficult' items, only individuals with higher levels of SE would *strongly agree*.

The three main advantages of IRT are its (a) ability to generate invariant item characteristics, (b) item level analysis, and (c) measurement precision. Item Response Theory procedures generate invariant item characteristics. An example of an invariant item characteristic is that the item has the same level of difficulty regardless of the ability level of the respondent, which allows for a more valid comparison of the level of the construct across individuals or groups [20,21]. This overcomes CTT's shortcomings in relation to its reliance on proportion correct or mean scores to determine an individual's ability level [20,21]. For example, in CTT, two individuals who do not have the same ability level (level of SE) may obtain the same mean score by strongly agreeing with the same number of items, although those items may have different difficulty levels. Second, unlike CTT's scale level analysis, IRT procedures examine the psychometric characteristics for each item [18]. The third advantage of IRT is that it assesses the measurement precision of each item, for individuals across a continuum of abilities [18]. In other words, the reliability of each item may vary from other items and for individuals with different ability levels. For this reason, reliability coefficients are considered more precise in IRT analysis in comparison to CTT procedures [18]. Despite these advantages, IRT procedures are also associated with some drawbacks. First, IRT methods are founded upon relatively strict assumptions and sample size requirements. Item Response Theory also requires that each response option be endorsed by a certain proportion of respondents.

*Polytomous IRT*

Various IRT models are available for items with different types of response options. For example, a dichotomous IRT model can be applied to scales where there are correct/incorrect answers. On the other hand, polytomous IRT is typically used for items with a Likert-type response scale (e.g., *strongly disagree* to *strongly agree*) where there are no correct or incorrect responses. Item characteristics are known as item parameters and individual ability levels are referred to as the latent trait (denoted as θ). In polytomous models, item parameters such as the item's location with respect to the level of ease or difficulty (i.e., denoted as the *b* parameter) and the item's ability to differentiate between individuals with high and low ability levels (i.e., denoted as the *a* parameter) are estimated and illustrated using Item Response Curves (IRCs) [18]. Unlike dichotomous IRT models with one item difficulty parameter (b), polytomous models have multiple item difficulty parameters (b), depending on the number of response options. These are known as step difficulty parameters, and reflect the ability levels required to endorse higher versus lower response options. In addition to item parameters, item and scale information curves, and item fit are also typically examined in IRT procedures.

The probability of selecting each response option is obtained for individuals on a continuum of ability levels and illustrated on an IRC. In polytomous IRT, the *b* parameter is defined as the point on the ability continuum of an IRC at which two consecutive response option curves intersect [19] (See b1–b4 in Figure 1). It is important to note that the step difficulty parameters should follow an order such that moving from a high response option to its next highest response option ought to be more difficult for respondents than moving from a lower response option to its next highest response option. For example, it is expected that moving from *strongly agree* (coded as 4) to *agree* (coded as 3) in CWEQ II would be more difficult (i.e., require higher level of SE) than moving from *agree* to *neutral* (coded as 2). This notion is also shown in Figure 1 where b1–b4 follow a consecutive order from lowest to highest. Also, if all of the category response curves of an item were located below or close to 0 on the continuum, this would indicate that the item was fairly easy. In the case of CWEQ II, this means even staff nurses with low SE levels would be likely to select the highest response option.

An item's ability to differentiate between individuals with high and low ability levels (i.e., denoted as the *a* parameter) is known as its 'discriminant ability'. For polytomous items, a highly discriminating item is one with a narrower and higher peaked response option curve which indicates that the response options differentiate between various ability levels fairly well [19]. Well discriminating items generally have a slope (a parameter) of greater than 1.0 [18].

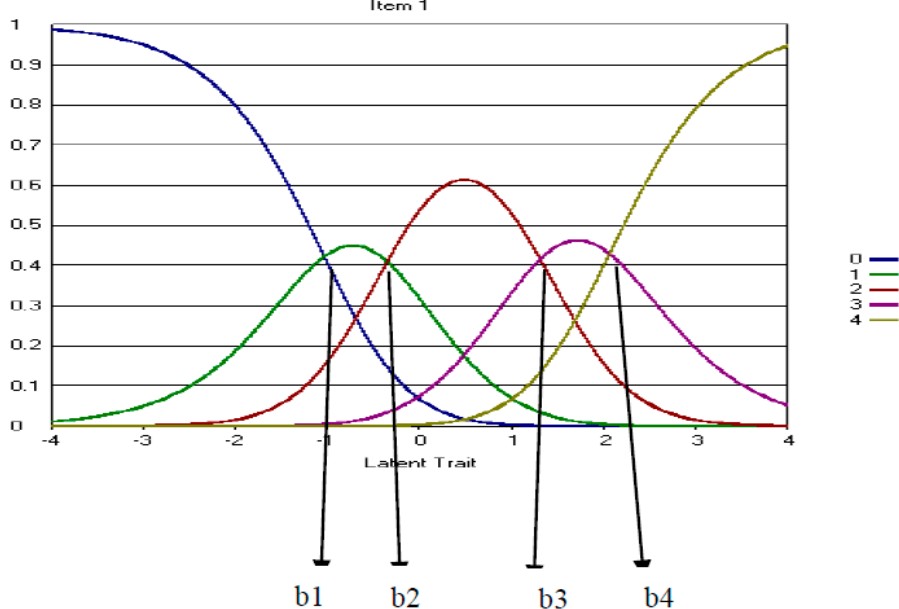

**Figure 1.** Step difficulty parameters on an item with five response options.

The information functions from each response option curve are combined to obtain an information function curve for the item. An item information curve demonstrates item reliability for a continuum of latent traits (SE levels). This is the latent trait level at which the item is most accurate as the amount of information provided by an item is inversely related to its standard error of measurement [18]. More discriminating items are typically known to provide more information about the latent trait, and therefore are more reliable [18]. Because there are no agreed-upon criteria to judge the adequacy of item information, Muraki recommended evaluating the graphical representation of item information curves as a measure of reliability [22]. Furthermore, item information curves are added to obtain a test information curve which demonstrates the overall reliability of the test for respondents on a continuum of ability levels [18].

Item fit depends on the consistency between respondents' actual scores on an item and the item scores predicted by the model [20]. The Q1 Chi square statistic and Z values are used as measures of item fit. For large sample sizes, Z values of greater than 4.6 indicate poor fit [23].

## 3. Methods

This study was a secondary analysis of data from a larger quasi-experimental study that investigated the effects of a nursing leadership development program on nurse leader and nursing staff outcomes. The one-year program was offered three times a year between 2007 and 2010 to novice nurse leaders who were selected by their Chief Nursing Officers to participate in the program. To determine the effectiveness of this program, data were gathered from the program participants and staff nurses whose leaders had attended the leadership development program. Data were gathered at baseline, prior to their nurse leaders beginning the program, and one year later, following the leaders' completion of the program. Surveys were mailed to nurse leaders who later distributed them among their staff. Only baseline data from staff nurses were used to examine the psychometric properties

of CWEQ II using IRT methods. Ethical approval for this research was obtained from the University of British Columbia Behavioural Ethics Review Board as well as the participating health authorities (H07-01559).

### 3.1. Sample and Sample Size Requirements

Baseline data from 1067 staff nurses were analyzed. Although there is no clearly defined sample-size requirement for IRT methods, the general rule of thumb is that sample size requirements increase as the model becomes more complex. For example, although a sample size of 100 may be adequate for parameter estimation in the most basic IRT model, known as Rasch models [24]; more complex models such as those required for polytomous scale analysis require a sample size greater than 500 [25]. Yen and Fitzpatrick similarly reported that a minimum sample size of 500 would be sufficient for accurate parameter estimation of a 25-item scale with five response options using polytomous IRT [20]. Based on this evidence, a sample size of 1067 nurses was deemed sufficient for this IRT study.

The majority of respondents were female (92%), with a mean age of 42 years (SD = 11). Sixty-three percent, 28%, and 9% of the staff nurses were working full-time, part-time and on a casual basis, respectively. In relation to nursing education, about 55% of the sample had a nursing degree (BSN or MSN) and 45% had a nursing diploma. Half of the nurses had 16 years or more of nursing experience. With respect to health care setting, 74% of the nurses worked in acute care settings followed by 23% and 3% that worked in Community Care and Long Term Care settings, respectively. A majority of the nursing staff worked in medical and/or surgical settings (26%) and in Emergency departments (13%).

### 3.2. Data Analysis

Descriptive statistics, exploratory factor analysis (EFA) and Polytomous IRT techniques were used to examine IRT assumptions and answer the study research questions. Descriptive statistics and EFA analyses were conducted using the Statistical Package for Social Sciences for Windows 20.0 (SPSS Inc., Chicago, IL, USA). For polytomous IRT, the 2-parameter partial credit model (2PPC) was selected to analyze this study's data using PARDUX software [20]. Estimates of item parameters and person latent trait for the 2-parameter partial credit model (2PPC) was provided using marginal maximum likelihood algorithms in PARDUX. Software known as IRT-lab was then used to generate IRCs, item and test information function curves, and item fit.

### 3.3. IRT Assumptions

IRT analysis is founded upon three assumptions: unidimensionality, local independence, and a minimum frequency of response option selection. First, when there is a fit between the scale and the applied measurement model in IRT, items are measuring only one underlying trait (i.e., are unidimensional) [26]. With respect to CWEQ II, unidimensionality means that the variance in staff nurses' item responses can be explained solely by their levels of SE. However, satisfying the strict assumption of unidimensionality is often impractical; therefore, this assumption may be replaced by the notion of "essential unidimensionality". Essentially unidimensional items may reflect several traits but one factor will very clearly dominate [26]. Similar to Yen and Edwardson [26], Slocum-Gori and Zumbo suggested the ratio of first to second Eigen values obtained through EFA as criteria for meeting this assumption [27]. A first- to second Eigen value ratio of greater than three was used as evidence of essential unidimensionality [26,27].

Local independence reflects the absence of correlation among items except through their relationship with the latent trait [26,28]. In other words, in case of CWEQ II, if SE were kept constant, an absence of correlation between items would indicate local independence. This also means that item residuals or error terms would be uncorrelated. Some common causes of local dependence include: respondent fatigue, external assistance/interference, and closely-related item content [29]. Local independence is usually evaluated using Q3 statistics, the correlation between any pairs of item

residuals after the latent trait, in this case SE, is kept constant [29,30]. According to Yen, Q3 statistics greater than .2 are evidence that the assumption of local independence has not been met [29].

The third assumption is related to the frequency with which each response option is selected. Edelen and Reeve noted that parameter estimation is adversely affected if certain response options are infrequently selected (i.e., selected by 5% or less of respondents) [31]. In such situations, response options are collapsed to increase the proportion of response option selection. Lecointe demonstrated that low-frequency categories can be combined with neighboring categories without seriously impacting item level reliability [32]. Wakita and colleagues compared three sets of questionnaires that were comprised of the same items but included various numbers of response options (i.e., four, five, and seven) [33]. They noted that participants tended to select more negative responses and avoided selecting response options at each end of the seven-point Likert-type scale in comparison to the three- or five- point scale [33]. Although reliability coefficients were independent of the number of response options, item values were negatively affected by a greater number of response options.

## 4. Results

Descriptive statistics are presented for CWEQ II items in Table 1. Overall, participants scored more items favorably than unfavorably, with 10 of the 19 items having a mean greater than 2.0 (where 0 = *strongly disagree*, 2 = *neutral*, 4 = *strongly agree*). Item means ranged from 1.63 to 3.05 with standard deviations ranging from 0.84 to 1.12.

### 4.1. Model Assumptions

Assumptions of essential unidimensionality and local independence were sufficiently met. Exploratory factor analysis using principal components analysis with a varimax rotation produced a five-factor solution explaining 65% of the variance in the items. This is inconsistent with the expected factor structure for the CWEQ-II, given its six subscales. Some items demonstrated inappropriate factor loadings. For example, item 14 from the informal power subscale loaded equally on two factors (Table 2). However, the conditions for essential dimensionality were met, as the ratio of first- to second Eigen values was 3.13, and all item loadings on the first factor were greater than 0.4. The scree plot (Figure 2) also illustrated the existence of a dominant factor. Therefore, the assumption of essential unidimensionality was sufficiently satisfied. Similarly, the assumption of local independence was adequately met as Q3 statistics of .2 or less were obtained for any pairs of items.

The frequency with which certain response options were selected was low for some items. The first response option (*strongly disagree*) was chosen 5% of the time or less for items 1, 2, 3 (opportunity subscale), 10, 11, (resources subscale), 13, 14 (informal power subscale), and 18 (formal power subscale). This may suggest a ceiling effect for these items. Similarly, their mean scores above 2.0 (*neutral*) may indicate that participants considered the items to be very easy. Similarly, response option 5 (*strongly agree*) was chosen less than 5% of the times for items 5, 6 (information subscale), 8 (support subscale), 10, 11 (resources subscale), 17, and 19 (formal power subscale) which may indicate item difficulty. Note that both extreme response options were selected infrequently for items 10 and 11. For this reason, we revised CWEQ II into a three-point response scale by collapsing the first two response options and the last two response options [32].

**Table 1.** Item Statistics: Means and Percentage of Responses.

| Items and Subscales | Mean (SD) | Strongly Disagree | Disagree | Neutral | Agree | Strongly Agree |
|---|---|---|---|---|---|---|
| 1. Challenging work (Opportunity) | 3.05 (0.89) | 0.8 | 3.0 | 23.4 | 35.7 | 37.1 |
| 2. Gain New Skills (Opportunity) | 2.80 (1.00) | 1.9 | 8.0 | 26.4 | 35.2 | 28.5 |
| 3. Tasks using all skills/knowledge (Opportunity) | 2.92 (0.94) | 1.3 | 6.2 | 21.9 | 40.3 | 30.3 |
| 4. Current State of hospital (Information) | 2.03 (1.01) | 7.5 | 18.9 | 44.5 | 21.2 | 7.9 |
| 5. Values of management (Information) | 1.68 (1.05) | 15.5 | 25.7 | 39.2 | 14.9 | 4.7 |
| 6. Goals of management (Information) | 1.63 (1.08) | 17.1 | 27.3 | 36.0 | 14.7 | 4.9 |
| 7. Support to information (Support) | 2.00 (1.06) | 10.8 | 17.7 | 38.8 | 26.7 | 6.0 |
| 8. Information about things to improve (Support) | 1.88 (1.02) | 11.0 | 21.4 | 40.0 | 23.5 | 4.2 |
| 9. Problem solving advice (Support) | 2.20 (1.07) | 8.1 | 14.7 | 35.8 | 31.8 | 9.6 |
| 10. Time to do paperwork (Resources) | 2.08 (0.92) | 3.8 | 22.6 | 40.1 | 28.9 | 4.6 |
| 11. Time to do job requirement (Resources) | 2.30 (0.88) | 2.0 | 22.6 | 40.1 | 28.9 | 4.6 |
| 12. acquiring temp help (Resources) | 1.86 (1.08) | 12.4 | 24.3 | 34.1 | 23.8 | 5.4 |
| 13. Collaborating on patient care (Informal Power) | 2.51 (1.10) | 4.1 | 13.5 | 32.5 | 27.4 | 22.5 |
| 14. Being sought by peers (Informal Power) | 3.01 (0.84) | 0.3 | 3.5 | 22.8 | 41.6 | 31.9 |
| 15. Being sought by management (Informal Power) | 1.97 (1.12) | 10.9 | 22.6 | 33.4 | 24.4 | 8.6 |
| 16. Seeking out other professionals' ideas (Informal Power) | 2.51 (1.10) | 5.3 | 12.2 | 28.7 | 34.0 | 19.8 |
| 17. Rewards for innovation (Formal Power) | 1.68 (1.05) | 15.7 | 25.4 | 38.1 | 17.2 | 3.6 |
| 18. Amount of flexibility (Formal Power) | 2.19 (0.98) | 4.6) | 18.1 | 39.7 | 28.7 | 8.9 |
| 19. Amount of visibility (Formal Power) | 1.8 (0.97) | 8.1 | 26.9 | 41.7 | 18.9 | 4.4 |

**Table 2.** CWEQ II Factor Loadings.

| Items (Subscales) | Factors | | | | |
|---|---|---|---|---|---|
| | **1** | **2** | **3** | **4** | **5** |
| 1. Challenging work (Opportunity) | 0.411 | 0.664 | | | |
| 2. Gain new skills (Opportunity) | 0.575 | 0.475 | | | |
| 3. Tasks that use all skills/ knowledge (Opportunity) | 0.475 | 0.593 | | | |
| 4. Current state of hospital (Information) | 0.544 | | −0.470 | | |
| 5. Values of top management (Information) | 0.649 | | −0.591 | | |
| 6. Goals of top management (Information) | 0.647 | | −0.577 | | |
| 7. Support to information (Support) | 0.751 | | | | |
| 8. Information about things to improve (Support) | 0.683 | | | | |
| 9. Problem solving advice (Support) | 0.713 | | | | |
| 10. Time to do paperwork (Resources) | 0.500 | −0.520 | | | |
| 11. Time to do job requirements (Resources) | 0.485 | −0.545 | 0.406 | | |
| 12. Acquiring temporary help (Resources) | 0.481 | | | | |
| 13. Collaborating on patient care (Informal Power) | 0.470 | | | 0.512 | |
| 14. Being sought by peers (Informal Power) | 0.489 | | | 0.489 | |
| 15. Being sought by managers (Informal Power) | 0.582 | | | | −0.421 |
| 16. Seeking out other professionals' ideas (Informal Power) | 0.469 | | | | |
| 17. Rewards for innovation (Formal Power) | 0.682 | | | | |
| 18. Amount of flexibility (Formal Power) | 0.559 | | | | |
| 19. Amount of visibility (Formal Power) | 0.627 | | | | |

Notes: Factor Loadings were obtained using principle component analysis. Factor loadings < 0.4 are omitted from this table.

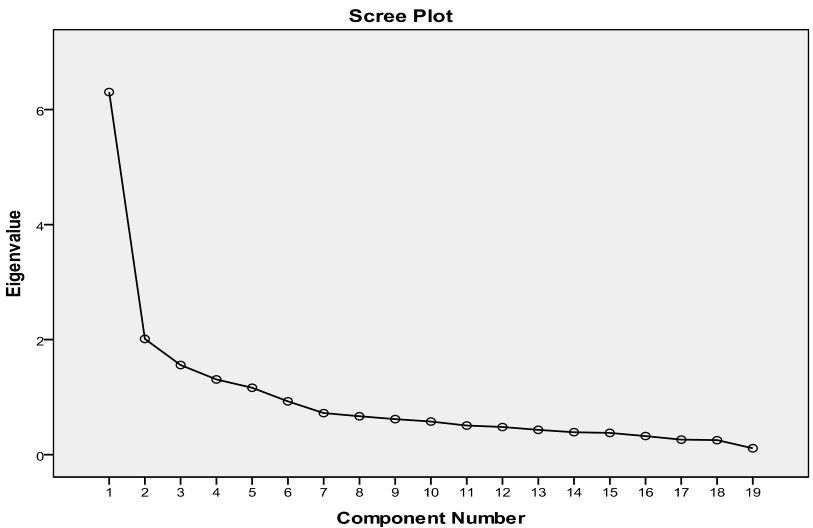

**Figure 2.** CWEQ II Factor Structure Scree Plot.

### 4.2. Five-Point CWEQ II

The sample's response patterns were used to estimate one discrimination parameter and four-step difficulty parameters per item (Table 3). Items with a slope parameter greater than 1.0 are generally considered as well-discriminating [18]. Only 4 of the 19 items discriminated well between respondents with high and low levels of SE; three items from the support subscale (items 7, 8, and 9) addressing *access to information support*, *information about things to improve*, and *problem solving advice* and one item from the informal power subscale (item 17) addressing *access to rewards for innovative behavior*. Although the lowest slope parameters were found for item 1 of the opportunity subscale (*a* = 0.47), and item 13 (*a* = 0.43) and 16 (*a* = 0.45) of the informal power subscale, the highest slope parameter belonged to item 7 (*a* = 1.73) of the support subscale. In other words, among all CEWE II items, item 7 was best able to differentiate between individuals with higher and lower levels of SE.

**Table 3.** Five-Point CWEQ II Item Parameter Estimates.

| Items and Subscales | Slope | SE | b1 | SE | b2 | SE | b3 | SE | b4 | SE |
|---|---|---|---|---|---|---|---|---|---|---|
| 1. Challenging work (Opportunity) | 0.47 | 0.40 | −1.89 | 0.19 | −2.35 | 0.09 | −0.53 | 0.09 | 0.05 | 0.08 |
| 2. Gain New Skills (Opportunity) | 0.72 | 0.27 | −2.32 | 0.14 | −1.68 | 0.08 | −0.41 | 0.08 | 0.47 | 0.09 |
| 3. Tasks using all skills/knowledge (Opportunity) | 0.53 | 0.30 | −2.11 | 0.15 | −1.58 | 0.08 | −0.71 | 0.08 | 0.42 | 0.08 |
| 4. Current State of hospital (Information) | 0.63 | 0.15 | −1.40 | 0.09 | −1.03 | 0.09 | 0.88 | 0.09 | 1.46 | 0.14 |
| 5. Values of management (Information) | 0.92 | 0.12 | −1.14 | 0.08 | −0.52 | 0.10 | 1.39 | 0.10 | 2.18 | 0.19 |
| 6. Goals of management (Information) | 0.88 | 0.11 | −1.03 | 0.08 | −0.35 | 0.11 | 1.32 | 0.11 | 2.07 | 0.18 |
| 7. Support to information (Support) | 1.73 | 0.17 | −2.30 | 0.11 | −1.41 | 0.10 | 0.93 | 0.10 | 3.50 | 0.20 |
| 8. Information about things to improve (Support) | 1.34 | 0.15 | −1.91 | 0.09 | −.98 | 0.10 | 1.06 | 0.10 | 3.34 | 0.21 |
| 9. Problem solving advice (Support) | 1.36 | 0.18 | −2.10 | 0.11 | −1.53 | 0.08 | 0.34 | 0.08 | 2.43 | 0.15 |
| 10. Time to do paperwork (Resources) | 0.60 | 0.18 | −2.26 | 0.09 | −0.75 | 0.08 | 0.45 | 0.08 | 2.28 | 0.17 |
| 11. Time to do job requirement (Resources) | 0.62 | 0.24 | −2.68 | 0.10 | −1.06 | 0.07 | 0.01 | 0.07 | 2.24 | 0.15 |
| 12. acquiring temp help (Resources) | 0.52 | 0.12 | −.96 | 0.08 | −0.42 | 0.09 | 0.49 | 0.09 | 1.86 | 0.16 |
| 13. Collaborating on patient care (Informal Power) | 0.43 | 0.18 | −1.51 | 0.11 | −1.04 | 0.08 | 0.17 | 0.08 | 0.36 | 0.09 |
| 14. Being sought by peers (Informal Power) | 0.59 | 0.61 | −3.22 | 0.19 | −2.31 | 0.09 | −0.75 | 0.09 | 0.41 | 0.08 |
| 15. Being sought by management (Informal Power) | 0.67 | 0.13 | −1.20 | 0.09 | −0.55 | 0.09 | 0.48 | 0.09 | 1.56 | 0.14 |
| 16. Seeking out other professionals' ideas (Informal Power) | 0.45 | 0.17 | −1.18 | 0.11 | −1.03 | 0.08 | −0.18 | 0.08 | 0.71 | 0.09 |
| 17. Rewards for innovation (Formal Power) | 1.09 | 0.13 | −1.28 | 0.09 | −0.53 | 0.11 | 1.34 | 0.11 | 2.92 | 0.22 |
| 18. Amount of flexibility (Formal Power) | 0.73 | 0.18 | −2.04 | 0.10 | −1.05 | 0.08 | 0.44 | 0.08 | 1.71 | 0.14 |
| 19. Amount of visibility (Formal Power) | 0.91 | 0.15 | −1.95 | 0.08 | −0.63 | 0.10 | 1.14 | 0.10 | 2.43 | 0.19 |

Item response curves were examined for each item. Item response curves illustrate the relationship between the probability of a staff nurse's selection of one particular response option and their level of SE. For items 1 and 16, the probability of selecting the second response option (i.e., *disagree*) is never higher than the probability of selecting the remaining response options (see Appendix A.1 Figure A1 for Item 1 IRC). Moreover, the probability of selecting the fourth response option (i.e., *agree*) is higher than the probability of selecting the remaining response options only for a narrow range of SE ($0 > \theta < 0.5$). Other CWEQ II items followed a similar pattern, where the probability of selecting the first and/or the fourth response options were higher only for a very narrow range of SE levels. These items include items 1, 3 (opportunity subscale), 4 (information subscale), 12 (resources subscale), 13, and 16 (informal power subscale). The remaining 13 IRCs illustrated that the probabililties of selecting each of the five response options were higher for a distinctly wider range of SE levels (Appendix A.1 Figure A2).

Step difficulty parameters and IRCs also provided an indication about the overall ease and difficulty of CWEQ II items. The IRCs for items 1, 2, 3 (opportunity subscale), 4 (Information subscale), 13, 14, and 16 (informal power subscale) as well as their b4 values indicated these were the easiest CWEQ II items. Items 7 and 8 of the support subscale were identified as the most difficult items. Furthermore, disordinal step difficulty paramters was noted for item 1where b2 is smaller than b1 (Table 3); moving from the second to the third response option (b2) required a lower level of SE than moving from the first to second response option (b1) (Appendix A.1 Figure A1).

A review of item information functions demonstrated that all items provided little information at extreme levels of SE. Items 1, 2, 3 (opportunity subscale), 14, 15, and 16 (informal power subscale) were identified as providing more information; therefore measuring lower SE levels with more accuracy or less measurement error (See Appendix A.1 Figure A3 for Item 1 Information Function). Other items provided more measurement precision at higher SE levels. For example, although the information curve for item 1 peaked at ability levels ranging from $-3$ to $-1$ (Appendix A.1 Figure A3), the information curve for item 18 peaked for ability levels ranging between $-2$ to $+1.5$ (Appendix A.1 Figure A4). Overall, the test information function showed that the five-point CWEQ II was most reliable for lower SE levels (i.e., $-2 < \theta > 0$) (Appendix A.1 Figure A5).

The fit of the partial credit model to the data was examined at the item level of the CWEQ II. With the exception of item 9 ($Z = 5.04$), a support subscale item, all CWEQ II items exhibited acceptable fit to the model (Table 4) with Z values less than 4.6.

**Table 4.** Five-Point CWEQ II Model Fit.

| Items and Subscales | Chi Square | DF | N | Z (fit) |
|---|---|---|---|---|
| 1. Challenging work (Opportunity) | 31.47 | 35 | 1056 | −0.42 |
| 2. Gain New Skills (Opportunity) | 27.66 | 35 | 1057 | −0.88 |
| 3. Tasks using all skills/knowledge (Opportunity) | 23.94 | 35 | 1057 | −1.32 |
| 4. Current State of hospital (Information) | 27.91 | 35 | 1038 | −0.85 |
| 5. Values of management (Information) | 22.75 | 35 | 1054 | −1.46 |
| 6. Goals of management (Information) | 17.90 | 35 | 1055 | −2.04 |
| 7. Support to information (Support) | 28.69 | 35 | 1058 | −0.75 |
| 8. Information about things to improve (Support) | 37.67 | 35 | 1056 | 0.32 |
| 9. Problem solving advice (Support) | 77.17 | 35 | 1057 | 5.04 |
| 10. Time to do paperwork (Resources) | 38.57 | 35 | 1059 | 0.43 |
| 11. Time to do job requirement (Resources) | 27.20 | 35 | 1058 | −0.93 |
| 12. acquiring temp help (Resources) | 54.12 | 35 | 1054 | 2.29 |
| 13. Collaborating on patient care (Informal Power) | 46.78 | 35 | 1025 | 1.41 |
| 14. Being sought by peers (Informal Power) | 38.11 | 35 | 1032 | 0.37 |
| 15. Being sought by management (Informal Power) | 48.88 | 35 | 1032 | 1.66 |
| 16. Seeking out other professionals' ideas (Informal Power) | 59.73 | 35 | 1028 | 2.96 |
| 17. Rewards for innovation (Formal Power) | 42.66 | 35 | 1024 | 0.92 |
| 18. Amount of flexibility (Formal Power) | 32.34 | 35 | 1027 | −0.32 |
| 19. Amount of visibility (Formal Power) | 68.84 | 35 | 1022 | 4.04 |

### 4.3. Three-Point CWEQ II

The infrequently selected response options for the original five-point CEWQ II served as an indication that fewer response options may be needed. As item parameter estimates are adversely affected by infrequent responses in a particular category [31], the first and second response options, and the fourth and fifth response options, were collapsed, producing a three-point response scale for the CWEQ II [32]. The factor structure and the Q3 statistics for the collapsed CWEQ II were similar to results for the five-point CWEQ II, supporting essential unidimensionality and local independence, and therefore, are not reported here.

The sample's response patterns were used to estimate one discrimination parameter and two-step difficulty parameters per item for the three-point CWEQ II. These parameter estimates for the three-point CWEQ II are presented in Table 5. After CWEQ II response options were collapsed, 7 of the 19 items (in comparison to 4 of the 19 items for five-point CWEQ II) discriminated well between respondents with high and low levels of SE, as indicated by their slopes greater than 1. These items were the two information items addressing *values and goals of management* (item 5 and 6), all support items (items 7, 8, 9), and two formal power items addressing *access to rewards for innovation and visibility* (items 17, 19). Item 2, from the opportunity subscale, concerned with *gaining new skills*, had a slope of 0.97. Including this item, 8 of the 19 items discriminated well between high and low levels of SE after response options were collapsed. Overall, slopes ranged from .61 to 2.07, with item 7 of the support subscale identified as having the most discriminating ability. Item 12 of the resource subscale, concerned with *acquiring temporary help*; and item 16 of the informal power subscale, addressing *seeking other professionals*, were identified as having the lowest slopes (i.e., .61). Overall, all item slopes were higher for the three-point CWEQ II than the five-point CWEQ II.

**Table 5.** Three-Point CWEQ II Item Parameter Estimates.

| Items | Slope | SE | b1 | SE | b2 | SE |
|---|---|---|---|---|---|---|
| 1. Challenging work (Opportunity) | 0.86 | 0.07 | −2.57 | 0.21 | −1.26 | 0.08 |
| 2. Gain New Skills (Opportunity) | 0.97 | 0.07 | −1.71 | 0.15 | −0.92 | 0.08 |
| 3. Tasks using all skills/knowledge (Opportunity) | 0.73 | 0.06 | −1.57 | 0.15 | −1.23 | 0.08 |
| 4. Current State of hospital (Information) | 0.87 | 0.06 | −0.83 | 0.09 | 0.70 | 0.08 |
| 5. Values of management (Information) | 1.22 | 0.07 | −0.22 | 0.08 | 1.38 | 0.11 |
| 6. Goals of management (Information) | 1.18 | 0.07 | −0.01 | 0.08 | 1.28 | 0.11 |
| 7. Support to information (Support) | 2.07 | 0.11 | −1.31 | 0.11 | 1.00 | 0.11 |
| 8. Information about things to improve (Support) | 1.64 | 0.09 | −0.83 | 0.10 | 1.14 | 0.11 |
| 9. Problem solving advice (Support) | 1.65 | 0.09 | −1.41 | 0.12 | −0.27 | 0.09 |
| 10. Time to do paperwork (Resources) | 0.65 | 0.05 | −0.62 | 0.08 | 0.34 | 0.08 |
| 11. Time to do job requirement (Resources) | 0.64 | 0.05 | −0.96 | 0.10 | −0.11 | 0.07 |
| 12. acquiring temp help (Resources) | 0.61 | 0.05 | −0.06 | 0.08 | 0.34 | 0.08 |
| 13. Collaborating on patient care (Informal Power) | 0.62 | 0.05 | −0.89 | 0.10 | −0.37 | 0.07 |
| 14. Being sought by peers (Informal Power) | 0.92 | 0.08 | −2.64 | 0.21 | −1.31 | 0.08 |
| 15. Being sought by management (Informal Power) | 0.85 | 0.06 | −0.26 | 0.09 | 0.29 | 0.09 |
| 16. Seeking out other professionals' ideas (Informal Power) | 0.61 | 0.05 | −0.78 | 0.11 | −0.58 | 0.08 |
| 17. Rewards for innovation (Formal Power) | 1.42 | 0.08 | −0.26 | 0.08 | 1.44 | 0.12 |
| 18. Amount of flexibility (Formal Power) | 0.92 | 0.06 | −0.97 | 0.10 | 0.28 | 0.08 |
| 19. Amount of visibility (Formal Power) | 1.13 | 0.07 | −0.50 | 0.08 | 1.12 | 0.10 |

Examination of IRCs and step difficulty paramters demonstrated that the step difficulty of the items decreased after response options were collapsed, as evidenced by b2 values of less than +1.5 for all items. This means that individuals with SE levels greater than +1.5 were most likely to select the third response option, suggesting that the scale could not differentiate between individuals with SE levels greater than +1.5. For example, an individual nurse with a SE level of +2 would be more likely to select the same response option than his/her counterpart with a SE level of +4. A similar situation holds at the lower SE levels as evidenced by b1 values greater than −1.5 for most items.

Therefore, the increase in discriminant ability is limited to a narrow range of SE levels, greater than −1.5 to less than +1.5. A comparison of IRCs in the Appendix demonstrates that IRCs improved for the problematic items noted above (e.g., item 1) (Appendix A.2 Figure A6), but worsened for other items; that is, items 3 (opportunity subscale), 12 (resources subscale), 13, 15, and 16 (informal power subscale). In Appendix A.2 Figure A6 you can also see that the disordinal step difficulty noted in item 1 from the original CWEQ II was resolved after response options were collapsed; moving from the first to second response option (b1 = −2.57) required lower levels of SE than moving from the second to third response option (b2 = −1.26) (Table 5). For item 18, the IRC shows a higher discriminant ability limited to a narrow range of SE levels (Appendix A.2 Figure A7).

A review of information functions demonstrated that collapsing response options led to slightly higher-peaked, but narrower, curves for several items (i.e., items 1, 6, 7, 14, 15, and 18) (Appendix A.2 Figures A8 and A9). The remaining items demonstrated loss of information. To determine whether collapsing response options improved reliability overall, test information curves for both versions of the CWEQ II were examined. These curves demonstrated that the five-point CWEQ II had a higher-peaked curve that covered a wider range of SE levels (Appendix A.1 Figure A5) in comparison to the three-point version (Appendix A.2 Figure A10). Thus, overall, the five-point CWEQ II was a reliable measure of SE for a wider spectrum of SE levels.

Assessment of item fit was conducted for the three-point CWEQ II. When response options were collapsed, the fit of item 9 improved. However, item 19, a formal power subscale item, demonstrated a poor fit with Z value of greater than 4.6. The remaining items demonstrated an acceptable fit to the model (Table 6).

**Table 6.** Three-Point CWEQ II Model Fit.

| Items and Subscales | Chi square | DF | N | Z (fit) |
|---|---|---|---|---|
| 1. Challenging work (Opportunity) | 14.66 | 17 | 1046 | −0.40 |
| 2. Gain New Skills (Opportunity) | 10.08 | 17 | 1047 | −1.19 |
| 3. Tasks using all skills/knowledge (Opportunity) | 17.63 | 17 | 1047 | 0.11 |
| 4. Current State of hospital (Information) | 13.34 | 17 | 1028 | −0.63 |
| 5. Values of management (Information) | 14.81 | 17 | 1044 | −0.37 |
| 6. Goals of management (Information) | 18.94 | 17 | 1045 | 0.33 |
| 7. Support to information (Support) | 16.58 | 17 | 1047 | −0.07 |
| 8. Information about things to improve (Support) | 18.85 | 17 | 1045 | 0.32 |
| 9. Problem solving advice (Support) | 24.42 | 17 | 1046 | 1.27 |
| 10. Time to do paperwork (Resources) | 7.79 | 17 | 1048 | −1.58 |
| 11. Time to do job requirement (Resources) | 19.10 | 17 | 1047 | 0.36 |
| 12. acquiring temp help (Resources) | 12.65 | 17 | 1043 | −0.75 |
| 13. Collaborating on patient care (Informal Power) | 13.87 | 17 | 1012 | −0.54 |
| 14. Being sought by peers (Informal Power) | 17.40 | 17 | 1019 | 0.07 |
| 15. Being sought by management (Informal Power) | 17.03 | 17 | 1019 | 0.00 |
| 16. Seeking out other professionals' ideas (Informal Power) | 19.00 | 17 | 1015 | 0.34 |
| 17. Rewards for innovation (Formal Power) | 31.75 | 17 | 1011 | 2.53 |
| 18. Amount of flexibility (Formal Power) | 16.70 | 17 | 1014 | −0.05 |
| 19. Amount of visibility (Formal Power) | 49.04 | 17 | 1009 | 5.49 |

## 5. Discussion

The purpose of this study was to examine the psychometric properties of CWEQ II using IRT analysis. The three study research questions included (a) Does an IRT model fit SE data obtained from staff nurses? (b) Does evidence support reliable and valid interpretation of SE scores? and (c) How does validity and reliability change with different numbers of response options?

With respect to the first research question, we noted that the two versions of CWEQ II fit the SE data equally, as each version had only one poor fitting item. With respect to the second research question, we found that the validity and reliability were partially supported for both versions of the

CWEQ II. We noted that discriminant ability was poor for a majority of the five-point CWEQ II items. Also, item 1 demonstrated disordinal step difficulty parameters. Item reliability was supported for a relatively wider range of SE levels. With respect to the three-point CWEQ II, although the discriminant ability of items improved, it improved for a narrower range of SE levels. The disordinal step difficulty in item one of the five-point CWEQ II was resolved after response options were collapsed. This is expected as reversal in step difficulty parameters was found to originate from the infrequent selection of certain response options [34]. The three-point CWEQ II demonstrated increased reliabilility but only for a relatively narrow range of SE scores. With respect to the third research question, a detailed comparison of validity and reliability evidence for each version of the CWEQ II is conducted in the following section.

### 5.1. Data-Model Fit

The IRT results demonstrated that, overall, there was an equally acceptable fit between the staff nurses' data and the indicated model for both the five-point and three-point versions of CWEQ II. Although item 9 was identified as poorly fitting the data in the original CWEQ II, this item was retained in the analysis as its Z value was close to the acceptable fit criterion. This decision was further justified as items ought not to be deleted based solely on item statistics, but also based on consideration of the item content [35]. We believe that item 9 content, related to problem solving advice, reflects an important and unique aspect of access to support in one's work environment. Although the fit of item 9 improved after CWEQ II response options were collapsed, item 19 illustrated a poor fit in the three-point CWEQ II; but with a value close to the acceptable criterion. Yen and Fitzpatrick reported using the wrong IRT model is one potential reason for obtaining poorly fitting items [20]. This may be the case here as a partial credit model, most appropriate for unidimensional constructs, was adopted to fit potentially multidimensional data consistent with the five-factor structure found by EFA results in this study.

### 5.2. Validity and Reliability

A majority of the five-point CWEQ II items showed poor discriminant abilities and low step difficulties. These items were predominantly from the opportunity, resources and informal power subscales. Their poor discriminant abilities mean that CWEQ II items had insufficient strength in differentiating among high and low levels of SE; small changes in the SE levels do not change the probability of respondents selecting a particular response option. In addition, most items were identified as relatively easy because even individuals with lower levels of SE would *strongly agree* with the item. However, the discriminant ability of most items increased and difficulty levels decreased after response options were collapsed. This is contradictory to Muraki's finding that the discriminant ability of most items decreased as a result of collapsing response options [22]. Collapsing the response options also resolved the disordinal step difficulty noted in item one of the five-point CWEQ II and enhanced item precision of measurement. This may suggest that a three-point response option would have been more appropriate for this item, but this was not done to maintain consistency and face validity.

Item information function and standard error of measurement curves confirmed that a majority of items from both versions of the CWEQ II provided the most information and subsequently the most reliability at *lower* (but not the lowest) levels of SE. Although the three-point CWEQ II generally produced narrower item information curves in comparison to the five-point scale, the test information curve showed that, overall, the five-point CWEQ II is more reliable for a wider range of SE levels. This finding is consistent with previous literature that also found that measurement precision was lost once response options were collapsed [22].

In summary, we first acknowledge that validity is not a binary condition but is rather a matter of degree [36]. Second, unlike traditional views of validity that have described validity as a property of the measure itself, more contemporary views have described it as the extent to which empirical evidence supports the intended meaning of test scores for its proposed purpose [37,38]. In other

words, measurement scholars acknowledge "validity is about whether the inference one makes is appropriate, meaningful, and useful given the individual or sample with which one is dealing, and the context in which the test user and individual/sample are working [38] (pp. 220–221). Based on this description, we expect the degree of validity of a measure to vary by context and/or respondents characteristics [37,38]. The above evidence suggested that both versions of the CWEQ II are less valid and reliable measures when used with nurses who have high levels of SE than when used with nurses who have lower levels of SE (but not the lowest). Therefore, given that employees' perceptions of SE reflect the conditions of their workplace environments, CWEQ II may not be the most appropriate measure of SE among nurses who are working in more ideal workplace environments such as "magnet hospitals". Magnet hospitals derive their accreditation, in part, by providing staff with optimal access to empowering structures.

*5.3. Implications*

CWEQ II requires further refinement before it can measure a wide range of SE levels with an adequate degree of validity and reliability. We acknowledge that the above recommendation is founded upon the assumption that all workplace environments provide some degree of access to empowering workplace structures and consequently SE levels are not expected to stand on the far left of the continuum (e.g., $\theta < -3$). This is important because items' reliabilities were also poor at the extreme left of the SE continuum.

In situations where the workplace environment of interest is not known to researchers, thus precluding estimations of the overall levels of nurses' feelings of SE, the five-point CWEQ II may be a more valid and reliable measure of SE than the three-point version. However, the three-point CWEQ II provides a more valid and reliable interpretation of SE scores in situations where nurses present with a restricted range of SE levels. It is often impossible to estimate the range of the ability level (in this case SE) among the study sample prior to data analysis, but if that information is available to researchers, the three-point CWEQ II may be a better option due to its higher discriminant ability for a particular narrow range of SE levels. Due to its higher discriminant ability, the three-point CWEQ II may be a better tool to use for pre-post measures that aim to capture differences in SE levels of individuals over time or differences in SE levels between settings or groups of nurses. In short, given that perceptions of SE are shaped by environmental conditions, the appropriate use of each version of the scale depends on the work setting targeted. This finding is similar to Chang's conclusion that "the issue of selecting four- versus six-point scales may not be generally resolvable, but may rather depend on the empirical setting" [39] (p. 205).

**6. Conclusions**

Item Response Theory procedures in this study provided the opportunity to obtain psychometric information about CWEQ II items that could not have been acquired using CTT procedures. This study's findings suggest that both versions of CWEQ II need further refinement. Overall, we reached two conclusions. First, the five-point CWEQ II is a more valid and reliable measure of SE among staff nurses working in less than ideal work environment conditions or in situations where workplace conditions are unknown to researchers. Second, the three-point CWEQ II is a more valid and reliable measure of SE if the workplace conditions reflect a narrow range of SE levels among staff nurses. Overall, the appropriate use of each version of the scale depends on the conditions of the work setting targeted. Future research should examine the CWEQ II items and their psychometric properties using a multidimensional IRT model. If a multidimensional model fits the data better, there can be important implications for the current conceptualization and measurement of SE.

**Acknowledgments:** We acknowledge Maura MacPhee as the principal investigator of the primary study from which the data were drawn. We thank Kadriye Ercikan for her feedback on a preliminary draft of this manuscript.

**Author Contributions:** Havaei was the lead investigator on this study and was responsible for the conceptual and methodological design of the study, data analysis, and interpretation of findings. Havaei and Dahinten shared equally in the writing of the manuscript.

**Conflicts of Interest:** The authors declare no conflict of interest.

## Appendix A.

Item Response Curves and Item Information Functions for a selection of the original five-point and the revised three-point CWEQ II items are provided in Appendix.

*Appendix A.1. The Original Five-Point CWEQ II Items*

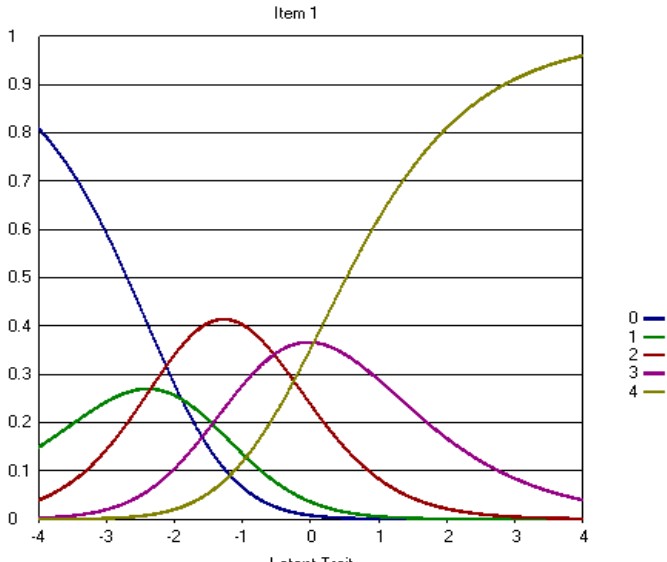

**Figure A1.** Item 1 IRC.

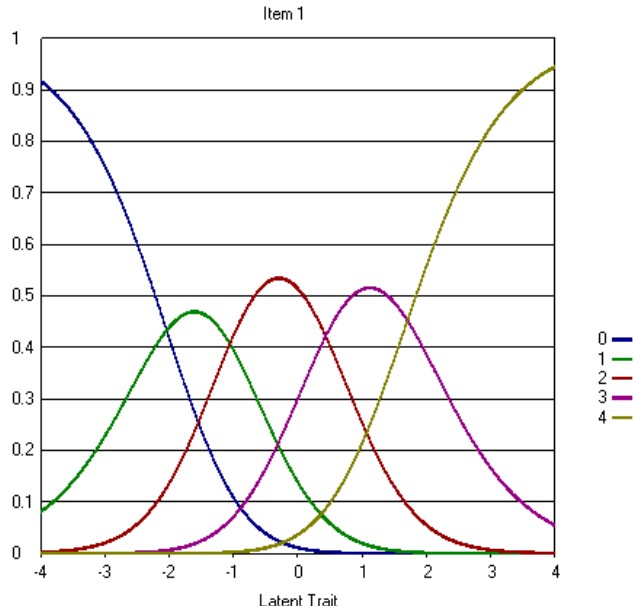

**Figure A2.** Item 18 IRC.

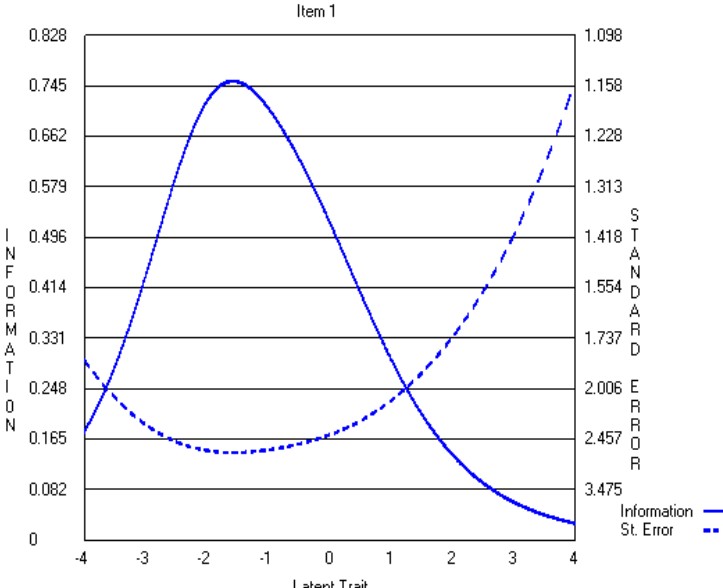

**Figure A3.** Item 1 Information Function.

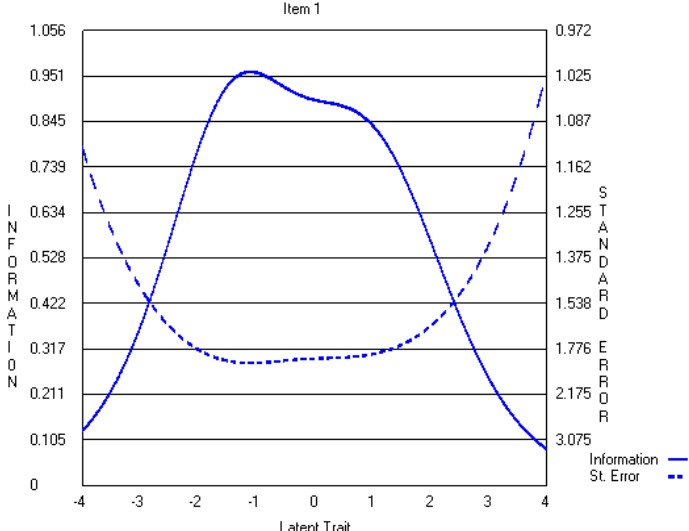

**Figure A4.** Item 18 Information Function.

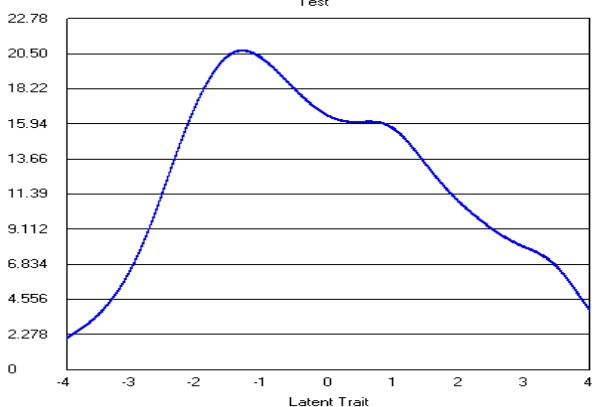

**Figure A5.** CWEQ II Test Information Function.

*Appendix A.2. The Revised Three-Point CWEQ II Items*

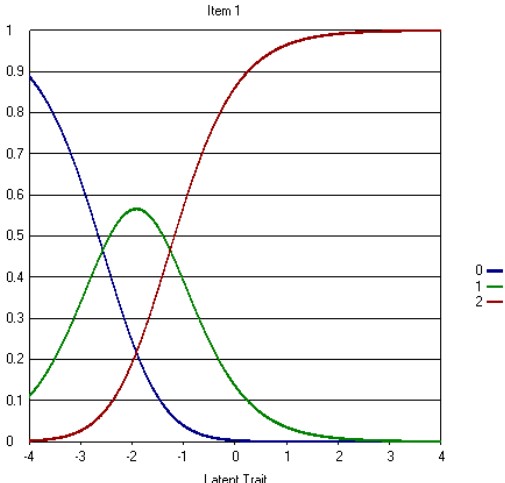

**Figure A6.** Item 1 IRC.

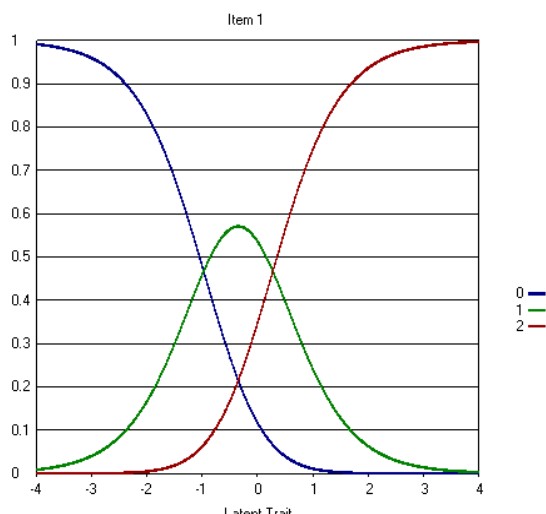

**Figure A7.** Item 18 IRC.

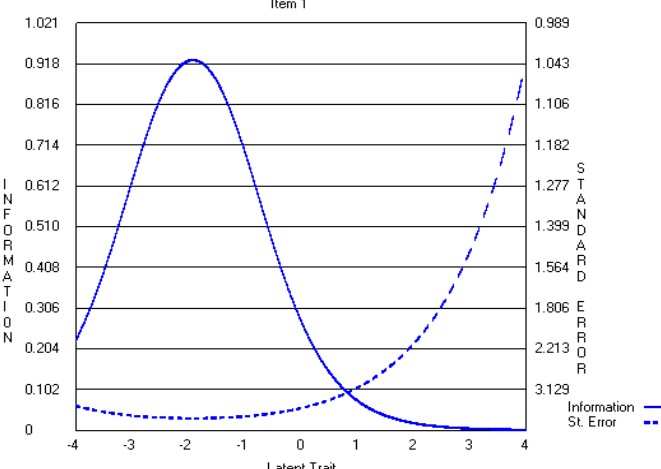

**Figure A8.** Item 1 Information Function.

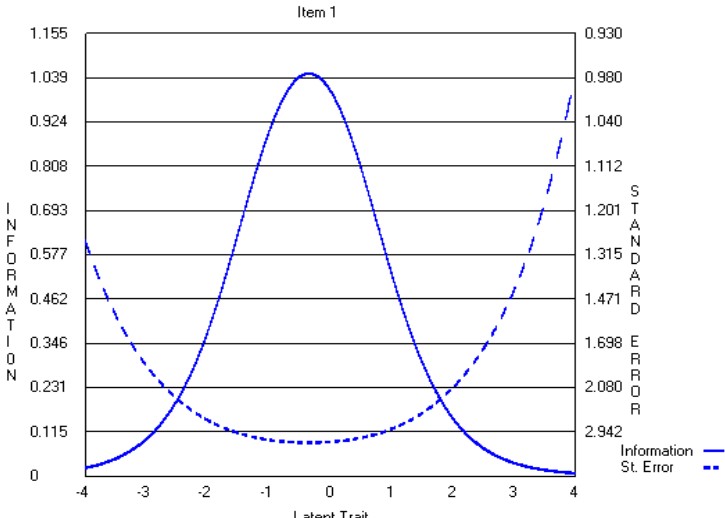

**Figure A9.** Item 18 Information Function.

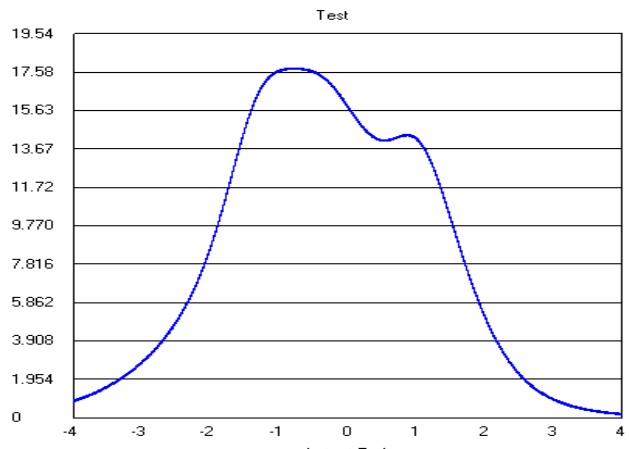

**Figure A10.** CWEQ II Test Information Function.

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
