# Peer review of "How Well Does the CWEQ II Measure Structural Empowerment? Findings from Applying Item Response Theory"

_admsci, doi:10.3390/admsci7020015_

Round 1

Reviewer 1 Report

The authors have contributed to extending our understanding of the psychometric properties of the CWEQ-2 by employing IRT. The paper is well written and clearly presented for readers who are new to this technique. There are a few minor comments provided to enhance the paper and to support the conclusion that each version of the CWEQ-2 are suited to different work setting conditions.

Page 5, paragraph 2: It would be useful to know more about the sample characteristics e.g. what settings did the nurses work in (acute, LTC, etc.); what type of units e.g. medical, surgical or specialty areas such as critical care or emergency? This data would be helpful given the assumption that different settings are associated with different levels of empowerment (based on other studies where the CWEQ-2 has been used).

Page16 (end of discussion section): In the discussion regarding when to use the 3 vs. 5-point scale, it would be helpful to expand the discussion further i.e. what recommendations would you suggest for pre-post measures to capture gains in SE? when trying to determine differences in SE between settings/groups of nurses where discrimination is more important to the study aims? It would be useful to also comment on the practical significance of the findings i.e. are the differences in scale scores of practical vs. scientific importance?

Page 16: Conclusion: The authors have suggested further refinement of both versions of the CWEQ-2.  It would be useful to recommend how this refinement could be approached. Do you recommend a different strategy for IRT given that the partial credit model was more appropriate for unidimensional constructs (as noted on page 15)? Are there other directions for future research you would suggest?

Minor edits:

Page 9, paragraph 2:  reduce spaces between ‘curves’ and ‘illustrate’

Page 9, paragraph 3: ‘parameters’ misspelled.

Page 14, paragraph 3: should read nurses’ data ( and not nurses data)

Author Response

Page 5, paragraph 2: It would be useful to know more   about the sample characteristics e.g. what settings did the nurses work in   (acute, LTC, etc.); what type of units e.g. medical, surgical or specialty   areas such as critical care or emergency? This data would be helpful given   the assumption that different settings are associated with different levels   of empowerment (based on other studies where the CWEQ-2 has been used).

Thank   you for this recommendation. In response, we have added the following   paragraph to the manuscript.

“With   respect to health care setting, 74% of the nurses worked in acute care   settings followed by 23% and 3% that worked in Community Care and Long Term   Care settings respectively. A majority of the nursing staff worked in medical   and/or surgical settings (26%) and in Emergency departments (13%).”

Page16 (end of discussion section): In the discussion   regarding when to use the 3 vs. 5-point scale, it would be helpful to expand   the discussion further i.e. what recommendations would you suggest for   pre-post measures to capture gains in SE? when trying to determine   differences in SE between settings/groups of nurses where discrimination is   more important to the study aims? It would be useful to also comment on the   practical significance of the findings i.e. are the differences in scale   scores of practical vs. scientific importance?

In   response to your feedback, we have added the following paragraph to the   manuscript.

“Due   to its higher discriminant ability, the three-point CWEQ II may be a better   tool to use for pre-post measures that aim to capture differences in SE   levels of individuals over time or differences in SE levels between settings   or groups of nurses.”

Page 16: Conclusion: The authors have suggested further   refinement of both versions of the CWEQ-2.  It would be useful to   recommend how this refinement could be approached. Do you recommend a   different strategy for IRT given that the partial credit model was more   appropriate for unidimensional constructs (as noted on page 15)? Are there   other directions for future research you would suggest?

In   response to your feedback, we have added the following paragraph to the   manuscript.

“Future   research should examine the CWEQ II items and their   psychometric properties using a multidimensional IRT model. If a   multidimensional model fits the data better, there can be important   implications for the current conceptualization   and measurement of SE.”

Page 9, paragraph 2:  reduce spaces between   ‘curves’ and ‘illustrate’

Thank   you for bringing this to our attention. This error is fixed now.

Page 9, paragraph 3: ‘parameters’ misspelled.

Thank   you for bringing this to our attention. This error is fixed now.

Page 14, paragraph 3: should read nurses’ data ( and   not nurses data)

Thank   you for bringing this to our attention. This error is fixed now.

Reviewer 2 Report

The CWEQ-2, a questionnaire designed to measure structural empowerment, has been widely used in nursing to test predictions about the causes and consequences of structural empowerment.  Using classical test theory, it has been found to be reliable and valid; a large body of evidence exists to show that people who have access to empowerment, as measured by the CWEQ2, are better off on a variety of dimensions. 

Until this paper, to my knowledge, it had not been examined using item-response theory.  The standards to show that a scale meets the criteria set item response theory are high and despite it being around for a number of years it is not routinely used in questionnaire development.  One of the problems with using this approach is that it requires a large sample size; this is not a problem with the current paper which tests the CWEQ using responses from 1067 nurses.  

This paper provides an excellent overview for people unfamiliar with item-response theory.  It carefully explains the theory and assumptions, tests the assumptions, and then walks the reader through the analysis. 

Whether or not it is reasonable to expect each item to discriminate at each response possibility is debatable.  But nevertheless, this paper does provide insight into the measurement of the construct.  More importantly, it raises the possibility that a Likert scale with three-response options might be better to assess the construct than the traditional 5-items.  This is an interesting possibility and contrasts with recommendations from classical test theory that the number of response options isn’t that important.

Given how stringent item-response theory is, it is not surprising that the scale did not fare particularly well.  I appreciate the fact that the author was conservative in his/her conclusions but I also find them intriguing. Should the 3-point CWEQ be used in places where the range of empowerment is narrow, and the 5-point CWEQ be used in less than ideal conditions?  This is an interesting possibility that warrants further investigation both from an empowerment point of view and from a measurement point of view.  And how do you compare between studies when response options are different?

Thus, although I was rooting for the CWEQ to do better, I still think that results are encouraging and do not take away from the fact that CWEQ predicts.  I would use this paper in classes to demonstrate how to use item-response theory.

Small typo:  on the second paragraph of the discussion, second sentence, “we respect to” should be “with respect to”

Author Response

The CWEQ-2, a questionnaire designed to measure structural empowerment,   has been widely used in nursing to test predictions about the causes and   consequences of structural empowerment.  Using classical test theory, it   has been found to be reliable and valid; a large body of evidence exists to   show that people who have access to empowerment, as measured by the CWEQ2,   are better off on a variety of dimensions. 

Until this paper, to my knowledge, it had not been examined using   item-response theory.  The standards to show that a scale meets the   criteria set item response theory are high and despite it being around for a   number of years it is not routinely used in questionnaire development.    One of the problems with using this approach is that it requires a large   sample size; this is not a problem with the current paper which tests the   CWEQ using responses from 1067 nurses.  

This paper provides an excellent overview for people unfamiliar with   item-response theory.  It carefully explains the theory and assumptions,   tests the assumptions, and then walks the reader through the analysis. 

Whether or not it is reasonable to expect each item to discriminate at   each response possibility is debatable.  But nevertheless, this paper   does provide insight into the measurement of the construct.  More   importantly, it raises the possibility that a Likert scale with   three-response options might be better to assess the construct than the   traditional 5-items.  This is an interesting possibility and contrasts   with recommendations from classical test theory that the number of response   options isn’t that important.

Given how stringent item-response theory is, it is not surprising that   the scale did not fare particularly well.  I appreciate the fact that   the author was conservative in his/her conclusions but I also find them   intriguing. Should the 3-point CWEQ be used in places where the range of   empowerment is narrow, and the 5-point CWEQ be used in less than ideal   conditions?  This is an interesting possibility that warrants further   investigation both from an empowerment point of view and from a measurement   point of view.  And how do you compare between studies when response   options are different?

Thus, although I was rooting for the CWEQ to do better, I still think   that results are encouraging and do not take away from the fact that CWEQ   predicts.  I would use this paper in classes to demonstrate how to use   item-response theory.

We   really appreciate your feedback and the recognition that this is the first   application of IRT with CWEQ II. It is also great to learn that this   manuscript can be used for pedagogical purposes.

Small typo:  on the second paragraph of the discussion, second   sentence, “we respect to” should be “with respect to”

Thank   you for bringing this to our attention. This error is fixed now.